# Potential impacts of climate change on the productivity and soil carbon stocks of managed grasslands

**N. J. B. Puche[1,2], M. U. F. Kirschbaum[3], N. Viovy[4], Abad Chabbi[1,2] ***

**1** UMR ECOSYS Joint research unit INRAE, AgroParisTech, Université Paris-Saclay, Palaiseau Cedex, France, **2** URP3F, French National Research Institute for Agriculture, Food and Environment (INRAE), Poitou-Charentes, Lusignan, France, **3** Manaaki Whenua – Landcare Research, Palmerston North, New Zealand, **4** LSCE/IPSL, CEA-CNRS-UVSQ, Laboratoire des Sciences du Climat et de l'Environnement, Universite Paris-Saclay, Gif-sur-Yvette, Frances

* abad.chabbi@inrae.fr

**Data Availability Statement:** All relevant data are within the paper and its Supporting information files.

**Funding:** AC received funding principally by the "New Zealand Government - Livestock Research

## Abstract

Rain-fed pastoral systems are tightly connected to meteorological conditions. It is, therefore, likely that climate change, including changing atmospheric $CO_2$ concentration, temperature, precipitation and patterns of climate extremes, will greatly affect pastoral systems. However, exact impacts on the productivity and carbon dynamics of these systems are still poorly understood, particularly over longtime scales. The present study assesses the potential effects of future climatic conditions on productivity and soil organic carbon (SOC) stocks of mowed and rotationally grazed grasslands in France. We used the CenW ecosystem model to simulate carbon, water, and nitrogen cycles in response to changes in environmental drivers and management practices. We first evaluated model responses to individual changes in each key meteorological variable to get better insights into the role and importance of each individual variable. Then, we used 3 sets of meteorological variables corresponding to 3 Representative Concentration Pathways (RCP 2.6, RCP 4.5 and RCP 8.5) for long-term model runs from 1975 to 2100. Finally, we used the same three RCPs to analyze the responses of modelled grasslands to extreme climate events. We found that increasing temperature slightly increased grasslands productivities but strongly reduced SOC stocks. A reduction in precipitation led to reductions of biomass and milk production but increased SOC. Conversely, doubling $CO_2$ concentration strongly increased biomass and milk production and marginally reduced SOC. These SOC trends were unexpected. They arose because both increasing precipitation and $CO_2$ increased photosynthetic carbon gain, but they had an even greater effect on the proportion of biomass that could be grazed. The amount of carbon remaining on site and able to contribute to SOC formation was actually reduced under both higher precipitation and $CO_2$. The simulations under the three RCPs indicated that grassland productivity was increased, but that required higher N fertilizer application rates and also led to substantial SOC losses. We thus conclude that, while milk productivity may continue at current rates under climate change, or even increase slightly, there could be some soil C losses over the 21st century. In addition, under the highest-emission scenario, the increasing importance of extreme climate conditions (heat waves and

Group of the Global Research Alliance on Agricultural Greenhouse Gases under grants SOW12-GPLER-LCR-PM (Proposal ID 16949-15 LCR), the AnaEE France (ANR, grant number 11-INBS-0001), the SOILWARM (ANR, grant number 21-CE01-0025), and the CarSolEI ADEME project 2252, (grant number 30001461). The funders had no role in study design, data collection and analysis, decision to publish, or preparation of the manuscript.

**Competing interests:** The authors have declared that no competing interests exist.

droughts) might render conditions at our site in some years as unsuitable for milk production. It highlights the importance of tailoring farming practices to achieve the dual goals of maintaining agricultural production while safeguarding soil C stocks.

## Introduction

Grassland ecosystems cover about one quarter of the Earth's total land area [1], with their soils containing large quantities of carbon. They are the primary source of fiber, meat and milk production, with important associated greenhouse gas emissions [2]. Vegetation dynamics drive C inputs to the soil but differ between ecosystems, primarily along climate gradients [3, 4]. There is therefore a strong correlation between soil organic carbon (SOC) stocks and temperature and precipitation [5]. Moreover, temperature and humidity also control SOC mineralisation and $CO_2$ fluxes [6] and thus the global and regional dynamics of SOC stocks.

Previous studies showed that SOC stocks of grasslands soils were sensitive to land use changes and management practices. Converting grassland to cropland generally led to SOC losses [7] and grassland degradation from overgrazing was also identified as detrimental to SOC stocks [8]. On the other hand, improved grassland management was reported to increase SOC stocks [9] with however large discrepancies between study sites, highlighting the uncertainties in the data related to management and the difficulties to generalize findings at large scale. Where soils have lost SOC due to agricultural practices, this has provided an opportunity for agricultural and managed grassland soils to contribute to climate change mitigation if these SOC losses can be reversed, leading to renewed interest in SOC management [10–13]. According to the 4p1000 international initiative, a small percentage increase (4 per mill) of global SOC stocks in the top 0.3–0.4 m of soils would be of a similar magnitude as the annual global net atmospheric $CO_2$ growth [14] so that a hypothetical increase of that magnitude could halt the growth of atmospheric $CO_2$. More recently, FAO (2019) initiated a new program aiming to help offset global anthropogenic $CO_2$ emissions by increasing the carbon storage capacity of soils. Both initiatives suggest that a moderate increase in global soil carbon stocks could offset a significant fraction of annual net anthropogenic CO2 emissions [15] and improve soil health while contributing to global food security [16]. However, climate change, introduces considerable uncertainty into the potential productivity of grasslands to meet the increasing global demand for food and possibly increase their carbon stocks in the soil.

With climate change, global surface temperatures are expected to increase over the next century together with greater spatial and temporal variability of rainfall patterns, and there may also be changes in the predictability of this variance [17–20]. Climate change has already begun to alter the Earth's climate system, with even larger changes expected over the next century, including more frequent droughts and heat waves with potentially serious threats to agroecosystems. These changes in meteorological conditions could have large impacts on carbon and water cycles as well as on productivity, phenology and biodiversity of terrestrial ecosystems [21, 22].

The adverse effects of some of these extreme changes are counteracted by higher $CO_2$ levels that are likely to benefit many plants through accelerated biomass production and improved water use efficiency [23, 24]. Agriculture is affected by climate change and, in turn, affects the climate through changes in greenhouse gas emissions from the land surface [25]. If there are any reductions in agricultural productivity, it will affect the food supply for an increasing global population and the need for other land to be brought into agricultural production to meet the overall food demand. There is, therefore, an urgent need (1) to improve predictions

of climate change impacts on agricultural productivity and soil carbon storage of agroecosystems; and (2) to assess whether agricultural productivity can be maintained at current levels; and (3) explore options and opportunities to adapt/modify management practices for optimal outcomes.

There is, therefore, a need to use models that are capable of projecting the potential impacts of climate change on long-term patterns of grasslands productivity and soil carbon stocks. Here, we used the CenW model to simulate carbon and water fluxes, net primary productivity (NPP), grasslands yields, soil respiration, milk production and changes of SOC stocks. We assessed how these variables would respond to changes in atmospheric $CO_2$ concentration, air temperature and rainfall. We also assessed the combined effects of precipitation, temperature and $CO_2$ changes on carbon dynamics under RCP 2.6, RCP 4.5 and RCP 8.5 [26] up to 2100. Our study focused on the impact of climatic variables and long-term agricultural management on SOC dynamics in the soil (up to 1 meter depth) and productivity of temperate managed grasslands in France. Specifically, our objectives were to: **(1)** assess the impacts of changes in key meteorological variables on modelled $CO_2$ fluxes, grassland and milk productivity and SOC stocks; and **(2)** simulate the potential influence of long-term future climate scenarios and climate extreme events on these variables.

## Materials and methods

### Experimental setup

The experimental site used for this modelling study was described in detail in previous studies [27–31]. Briefly, the site is located in Lusignan, Poitou-Charentes, southwestern France (46˚ 25′13′′ N; 0˚07′29′′ E) and is part of a long-term observatory for environmental research that was first launched in spring 2005. The mean air temperature and precipitation were 11.2˚C and 774 mm yr$^{-1}$, respectively, with no use of irrigation. The upper soil horizons are characterized by a loamy texture, classified as Cambisol, and the lower soil horizons have a clayey texture rich in kaolinite and iron oxides, classified as a Paleo-Ferralsol [32, 33]. The study site was under agricultural use for at least 100 years. Prior to the start of the observatory, the site was partly under permanent grassland and the other part under grassland-crop rotations. The whole site was then ploughed and after 2 years of homogenisation under mustard cultivation ("baseline" and/or "initilisation"), the agricultural treatments were implemented. The sown grasslands consisted of a mix of three grass species (*Lolium perenne*, *Festuca arundinacea* and *Dactylis glomerata* L.) and white clover (*Trifolium repens*). The 22.5 ha study area was divided into 20 small plots of 0.4 ha each (5 treatments with 4 replicates) and four larger plots of 3 ha each, to test different forcing variables: crop/grass rotation durations, levels of N application, and mowed versus grazed pastures. The details of the different treatments can be found in [30].

In the present study, we have selected two larger permanent grassland paddocks of 3 ha, both equipped with an eddy covariance measurement system and a meteorological station. One of the paddocks was regularly N fertilized and mowed with harvested hay exported off-site. Dairy cows regularly grazed the other paddock on a rotational basis, with all animal excreta directly returned to the paddock and with no additional application of manure. Nitrogen fertilizer was applied to the grazed paddock but at lower rates [31]. On average, the mowed paddock received (> 320 kg N ha$^{-1}$ with 3 to 4 cuts per year depending on climatic conditions. For the grazed paddock, the system receives on average <50 kg N ha$^{-1}$ (not including nitrogen returned in dung and urine during grazing) and is grazed 7.4 times per year with grazing events spread, on average, over 5 consecutive days and a stocking rate of 17 heads per ha. Meteorological conditions for the model parameterization and validation over the two

selected paddocks were acquired from a weather station placed 1.9 m above the soil surface on the mown paddock, coupled to a data logger (CR-10X, Campbell Scientific Inc.), as described by [29].

## Modelling details

**CenW ecosystem model overview.** We used the CenW ecosystem model [34] for our model-based study. CenW is a dynamic process-based model, combining the major carbon, energy, nutrient and water fluxes in an ecosystem [34]. CenW uses site characteristics, climate data and management input to predict how different pools, such as leaves, roots and different fractions of organic carbon, will change over time under the influence of any external factors, such as weather variables or management choices. For the present work, we used CenW version 4.2, which runs on a daily time step and dynamically simulates canopy photosynthesis and vegetation growth through the allocation of fixed carbon to leaves and roots. The carbon and nitrogen cycles are completed through litter fall, decomposition, autotrophic and heterotrophic respiration, and their relationships to climatic drivers, allowing the simulation of the behavior of ecosystems over time. Soil organic carbon dynamics modelled by CenW are based on the CENTURY model, which was originally developed for grasslands [35] and CenW has previously been applied successfully to the modelling of gas exchange of grassland systems [31, 36–38]. The model includes three soil organic carbon pools (active, slow and resistant) and two litter pools (metabolic and structural), with different potential decomposition rates. Leaf and root senescence and litter production are controlled by plant types and phenology, and by water, temperature and plant-specific senescence parameters.

**Simulation setup.** CenW was previously calibrated, parameterized and validated using daily-summed 30-minute fluxes of $CO_2$ and water measured by eddy covariance systems between 2006 and 2010 from paired, but differently managed (mowed and rotationally grazed), grassland sites [31]. Briefly, model simulations were optimized for the mowed and grazed paddocks by an automatic parameter optimization procedure imbedded in the model that aims to maximize the agreement between model and observations. As inputs, the CenW model used daily minimum and maximum air temperature, precipitation, absolute humidity, global solar radiation and atmospheric $CO_2$ concentration measured at the experimental site. To simulate net carbon fluxes in managed grasslands system effectively, it was essential to know the timing of grazing and harvesting carried out on the modelled paddocks [31, 36].

For the grazed paddock, the model assumed that cows consumed a fixed proportion of above ground biomass at each grazing event. Of that feed, 50% was assumed to be lost by respiration [39], 5% as methane [40], and 18% removed as milk solids [39, 41, 42], with the remaining 27% returned to the paddock in dung and urine. It was also assumed that animal weights remained constant and not added to carbon gains or losses from the paddocks.

For the mowed paddock, we assumed that during each harvest event, 80% of aboveground biomass was cut, and of that amount, 95% was exported from the paddock while 5% remained on the paddock as residue. Initial model conditions were set up through the spin up of the model until total soil carbon (up to 1 meter depth) reached equilibrium conditions that were consistent with the observed SOC measured in 2005. We used detailed farm records [31] to set details of the event management procedure in CenW that principally required the timing, added amounts of mineral nitrogen fertilizer applications and characteristics of all harvests/grazing events.

The optimized parameter files and initial conditions from [31] were used in all CenW simulations used for the present study for current and all future simulations under different aspects of climate change.

**Model sensitivity to meteorological conditions.** For the simulations under changed climate conditions, we standardized management practices and soil and vegetation conditions observed at the Lusignan study site between 2005 and 2010 [29, 31]. These standards were implemented through an automatic mineral N fertilizer-application procedure that triggered applications when a specified nitrogen limitation (0.8) was reached. Harvest and grazing events were also controlled by an automatic routine that triggered harvesting or grazing events whenever above ground biomass reached 2.1 and 1.8 tC ha$^{-1}$ for harvest and grazing events, respectively. Those procedures ensured that N fertilizer applications and grazing/harvests biomass removals were automatically adjusted in line with any changes in grasslands productivity caused by different meteorological conditions.

To ensure proper capture and representation of inter-annual variability of climatic conditions [37] we used the weather data measured over the period of 2001–2016 repeatedly (4 times) for longer model runs (64 years). To quantify the sensitivity to climatic conditions, we then ran various simulations, each time changing one of the key variables at a time, while maintaining the others at their base-line values.

For testing the effect of precipitation, we changed precipitation amounts from − 25% to + 25% at increments of 10% but did not explore the effects of changing seasonal precipitation patterns, or the frequency of droughts. Reduced precipitation was simulated by multiplying observed daily precipitation by the indicated fractions. To simulate increased precipitation, we added extra amounts of water every 7 days to achieve targeted annual totals with given percentage increases.

For the $CO_2$ sensitivity test, we changed $CO_2$ concentration from– 15% (331.5 ppm) to a doubling (780 ppm) of the reference level (390 ppm). For testing the effect of the temperature changes, we changed air temperature by– 4˚C,– 2˚C,– 1˚C, + 1˚C, + 2˚C and + 4˚C. To avoid the development of unrealistic vapor pressure deficits, we also recalculated absolute air humidity by adjusting dew point temperatures by the same amount as air temperatures [37], so essentially maintaining the same diurnal temperature range. For each of these sensitivity scenarios, we only changed one of the forcing variables at a time and kept all others unchanged (except for the temperature scenarios for which air humidity was also adjusted as described above). All simulations were run for 64 consecutive years to produce the grassland system's responses under different climate sensitivity scenarios. Practically, all meteorological variables were not changed for the first 32 years of simulation and were then abruptly changed to the desired value for the next 32 years. For these simulations, results are reported on an annual basis (1st January to 31 December) in tC ha$^{-1}$ yr$^{-1}$ except for nitrogen fertilizer application rates which are given in kgN ha$^{-1}$ yr$^{-1}$ and were averaged over the last 10 years of the simulation period. The mean is the average of the different annual variable values and the variability represent the year-to-year changes in modelled variables over those last 10 years.

**Climate change scenarios.** The set of RCPs were adopted for climate-change impact assessment studies for the IPCC's Fifth Assessment Report [43]. RCP 2.6 is the most optimistic scenario, characterized by very low GHG emissions with maximum radiative forcing reached before 2050 and sharp emission reductions after the peak emissions. RCP 4.5 is a stabilization scenario, where total radiative forcing is stabilized by 2100 using a range of strategies for reducing anthropogenic GHG emissions. The most pessimistic scenario is RCP 8.5, corresponding to high GHG concentrations and characterized by continuous increases of GHG emissions beyond 2100.

For this study, we selected data from the CNRM2014 experiment because all meteorological variables needed for running CenW were available at a fine spatial resolution of 8 km for all of France [44]. The CNRM2014 experiment consisted of the dynamic downscaling of modelled climate variables under different climate change scenarios from the global to regional (France)

scales. At first, atmospheric GHG concentrations from the different RCP scenarios were used to run the CNRM-CM5 v5.1 and ARPEGE-Climat v5.1 global circulation models. Then, results from these models were used to inform a regional climate model called ALADIN-Climat v5.2 that delivered continuous daily meteorological variables at a 12-km resolution across France. Finally, daily meteorological variables were projected onto an 8-km grid and bias corrected. The dataset was accessed through the DRIAS portal (https://drias-prod.meteo.fr/okapi/accueil/okapiWebDrias/index.jsp?lang=en) and included projections of daily precipitation (Figs 1 and 2e, 2f in S1 File), minimum and maximum air temperatures, air humidity and global solar radiation for the reference period (1971–2005), and for the period 2006–2100 for the RCP 2.6, RCP 4.5 and RCP 8.5 scenarios. The atmospheric $CO_2$ concentrations data under the different RCP scenarios were downloaded from the RCP Database version 2.0.5 (http://www.iiasa.ac.at/web-apps/tnt/RcpDb).

The total simulation period (1975–2100) was separated into four different time slices of about 30 years, each. These periods are hereafter referred to:

- Historical (H): 1975–2005

- Near future (NF): 2006–2036

- Intermediate future (IF): 2037–2067

- Far future (FF): 2068–2100

Table 1 summarizes projected average atmospheric $CO_2$ concentrations, air temperature, and annual precipitation for our study site for different time slices until the end of the 21st century under the 3 selected RCP scenarios. We also tested the importance of $CO_2$ fertilization on modeled SOC and grassland productivities. For this, we used the same daily meteorological variables simulated for the Lusignan site but maintained the $CO_2$ concentration in the atmosphere constant at the value reached at the end of the H period (330 ppm).

The more optimistic RCP 2.6 scenario resulted in the lowest atmospheric $CO_2$ concentration which would peak at 477 ppm in about 2050 (IF period) before falling back to 421 ppm by 2100 (Table 1 and Fig 2a, 2b in S1 File). Under RCP 4.5, the $CO_2$ concentration would increase faster and for longer than under RCP 2.6, but would nearly plateau within the FF period (Fig 2a in S1 File) and reach 538 ppm by 2100. Under RCP 8.5, $CO_2$ would continue to increase throughout the 21st century at a greater rate than those under RCP 2.6 and RCP 4.5 to reach 936 ppm by 2100 (Table 1 and Fig 2a, 2b in S1 File).

Observed annual air temperature has already been rising strongly even during the baseline period (Table 1 and Fig 2c, 2d in S1 File), and is expected to continue increasing under all the three scenarios (Fig 2c, 2d in S1 File). Averaged over the defined 30-year periods, historical mean air temperature was 11°C (Table 1 and Fig 2d in S1 File), and by the FF period, temperatures are expected to increase on average to 12.7, 13.3 and 15.1°C under RCP 2.6, RCP 4.5 and RCP 8.5, respectively (Table 1).

**Table 1.** Averages of projected annual climate variables (atmospheric $CO_2$ concentration, air temperature and annual precipitation) under the three selected RCP scenarios (RCP 2.6, RCP 4.5 and RCP 8.5) over the four different time slices: Historical (H), near future (NF), intermediate future (IF) and far future (FF).

| | | H | RCP 2.6 | | | RCP 4.5 | | | RCP 8.5 | | |
|---|---|---|---|---|---|---|---|---|---|---|---|
| Meteorological variable | units | H | NF | IF | FF | NF | IF | FF | NF | IF | FF |
| Atmospheric $CO_2$ | ppm | 353 | 412 | 441 | 429 | 414 | 490 | 532 | 422 | 557 | 795 |
| Air temperature | °C | 11.0 | 12.3 | 12.8 | 12.7 | 12.0 | 12.3 | 13.3 | 11.9 | 12.9 | 15.1 |
| Precipitation | mm yr$^{-1}$ | 776 | 869 | 851 | 884 | 862 | 853 | 752 | 837 | 792 | 754 |

Inter-annual as well as inter-scenario variations of annual precipitation amounts are very important (Table 1, and Fig 1 and 2 in S1 File). The average annual precipitation of the baseline period (H) was 776 mm yr$^{-1}$ (Table 1), and the projected precipitation showed a slight increase under all RCP scenarios during the NF and IF periods (Table 1 and Fig 2f in S1 File). Under the different RCPs, precipitation was projected to decrease only marginally in the different 30-year periods, with the trend change being small compared to normal inter-annual variability (Fig 2e, 2f in S1 File).

**Extreme climatic events.** The study of extreme climatic events due to global climate change has become increasingly important due to their significant impacts on natural processes. The Expert Team on Climate Change Detection and Indices (ETCCDI) has defined 27 climate change indices focusing on air temperature and precipitation extremes [45].

We selected 6 out of the 27 ETCCDI indices as the most relevant for our study region to quantify the temporal evolution of the frequency of temperature and precipitation extremes over the past (1975–2005) and future (2006–2100) periods. For this analysis, the R package "climdex.pcic" version 1.1–9.1 was applied to the ensemble of meteorological data used to run the CenW model for the different RCP scenarios with the 1975–2005 period selected as the reference. The first three selected indices relate to extreme hot temperatures and the last three to precipitation. The number of hot days (NHD) is calculated as the number of days in a year with air temperatures above 25°C, the second (TX90P) correspond to the proportion of days with temperatures above the 90th percentile of the reference period. The warm spell duration index (WSDI) characterizes the length of heat waves and is defined as the number of days in a year with at least 6 consecutive days with max temperature above the 90th percentile of the reference period. The maximum length of the dry spell index (MLDS) represents the annual maximum number of consecutive days where daily precipitation is less than 1 mm d$^{-1}$. The annual sum of precipitation in days where daily precipitation exceeds the 95th percentile of daily precipitation in the base period (R95pTOT) and represents extremes precipitation events. The last index, "prcpTOT", is the annual sum of precipitation in days with rainfall above 0.1 mm.

## Results

The two studied pastoral systems (i.e., grazing and mowing management) showed very similar responses to possible modeled climate change effects. The main text, therefore, includes only graphs for grazed pastures. Equivalents results for mowed pastures are shown in the Figs 3–10 in S1 File. Numeric information given in the text refers to grazed pasture while numbers in parentheses refer to the corresponding data for mowed pastures.

### Characteristics of extreme-climate indices for Lusignan

Table 2 shows the results of the extreme-climate indices calculated for Lusignan under the different RCP and time slices.

The different indices are expected to be quite different and display different trends under the different scenarios and periods. NHD is expected to increase for all scenarios compared to the H period and throughout the century, with rates depending on the RCP. In FF, for example, NHD is expected to increase by 71, 91 and 160% under RCP 2.6, 4.5 and 8.5, respectively.

TX90P is also expected to increase from H to FF at rates that depend on emissions scenarios. In the FF periods by the end of the 21st century, extremely hot days are expected to increase 2-fold under RCP 2.6, 2.5-fold under RCP 4.5 and 4-fold under RCP 8.5. WSDI is expected to increase for all periods, and under all RCPs compared to the historical climate period (Table 2), with continuous increases throughout the coming century. Not surprisingly,

**Table 2. Values of the 6 selected ETCCDI climate change indices used in this study for the Lusignan area.** The six indices are based on daily temperature and precipitation under the three different RCP scenarios and for the historical (H), near future (NF), intermediate future (IF) and far future (FF) periods.

| code | definition | units | H | RCP 2.6 | | | RCP 4.5 | | | RCP 8.5 | | |
|---|---|---|---|---|---|---|---|---|---|---|---|---|
| | | | H | NF | IF | FF | NF | IF | FF | NF | IF | FF |
| NHD | Number of hot days | d | 36 | 47 | 60 | 61 | 46 | 50 | 68 | 40 | 62 | 93 |
| TX90p | Percentage of days when $T_{max}>90^{th}$ percentile | % | 10 | 18 | 21 | 21 | 15 | 18 | 26 | 14 | 22 | 41 |
| WSDI | Warm spell duration index | d | 8 | 21 | 28 | 26 | 17 | 25 | 42 | 12 | 26 | 90 |
| MLDS | Maximum length of dry spell | d | 25 | 30 | 26 | 27 | 26 | 25 | 33 | 23 | 28 | 37 |
| R95pTOT | Annual total precipitation $>95^{th}$ percentile | mm yr$^{-1}$ | 155 | 244 | 232 | 271 | 195 | 207 | 191 | 190 | 205 | 215 |
| prcpTOT | Annual total precipitation when daily rainfall $>0.1$ mm | mm yr$^{-1}$ | 756 | 842 | 839 | 860 | 835 | 830 | 744 | 816 | 777 | 734 |

WSDI is expected to increase the most by the FF period under the most extreme emission scenario (RCP 8.5) with an increase from just 8 days in the H period to 90 days.

In the H period, MLDS was 25.2 d (Table 2), and is expected to remain nearly constant over all periods under RCP 2.6. The MLDS is also expected to remain stable throughout NF and IF under the other scenarios but in FF, it is expected to increase by 8 and 12 days under RCP 4.5 and RCP 8.5, respectively.

The R95pTOT index is expected to show different trends under different RCP scenarios but is expected to always be higher than for the H period. Interestingly, prcpTOT is expected to increase over all periods under RCP 2.6 but is expected to be reduced slightly in FF under both RCP 4.5 and RCP 8.5 (Table 2 and Fig 2 in S1 File).

**Model responses to factorial changes in climate variables.** Here, we separately show the modelled grassland systems responses to changes in temperature, precipitation and $CO_2$ concentration. In these sections, figures show the modelled responses of gross primary productivity (GPP), ecosystem respiration (ER), grasslands productivities, shown as net primary production (NPP), milk production (milk solids), mineral nitrogen fertilizer and SOC stocks in the grazed paddock to one-time changes in these meteorological variables. The mean responses shown below represent the average of annual variables over the last ten years of the simulation period, with the shown variability being the year-to-year variability in annual means.

*Effects of changes in air temperature*. Photosynthesis (GPP, Fig 1a), ecosystem respiration (ER, Fig 1b), net primary production (NPP, Fig 1c), and milk production (Milk solids, Fig 1d) were all positively correlated with changes in air temperature. In contrast, soil organic carbon stocks (SOC, Fig 1f) were negatively correlated with air temperature (i.e., SOC decreased with increased air temperature). However, there was no trend in the need for application of mineral N fertilizer (N fertilizer, Fig 1e).

GPP was generally not very sensitive to changes in air temperature except when mean air temperature in the simulations was 4°C lower than for the reference period when GPP was reduced by −9% (−8%, numbers between parentheses are for the **mowed** site for which the corresponding Figs. are in Figs 3–10 in S1 File. Inter annual variations of mean annual GPP for the different temperature change scenarios were also important (large whiskers) with ΔGPP (max-min GPP) of 10 tC ha$^{-1}$ yr$^{-1}$. ER responded almost linearly to changes in air temperature, the grazed (mowed) grassland ER varied from −11% (−7%) under T−4°C to +5% (+6%) under T+4°C. Under a 4°C reduction of air temperature, NPP was reduced by −20% (−18%) and increased by +7% (+8%) with an increase of air temperature of 4°C. Average milk-solid production experienced a sharp reduction of −34% (−33%) under T−4°C. Milk production was often marginal under the Lusignan climatic conditions, and in some years under reduced temperature scenarios, milk-solid production was reduced to zero. More specifically, that was

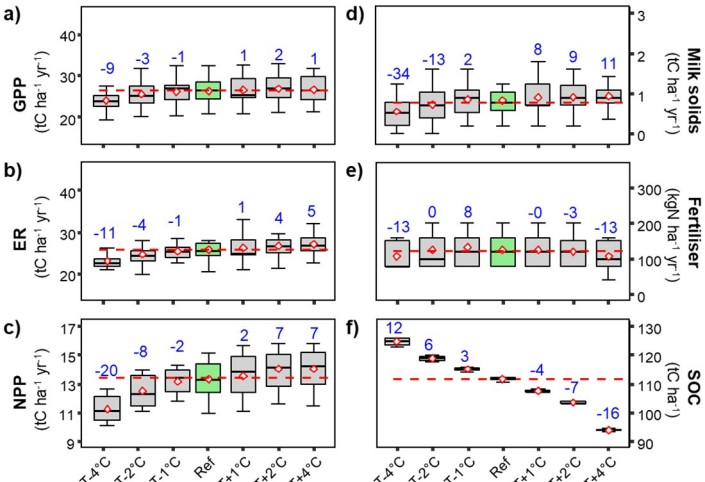

**Fig 1. Box and whiskers charts for modelled variables under a range of different temperatures.** Shown are modelled GPP a, ER b), NPP c), Milk-solid production d), required N fertilizer application rates), and SOC stocks f) in response to changes in air temperature for the grazed grassland site. Horizontal red dashed lines are the means of the variables for the reference period and blue numbers give the percentage changes of the variable means under the different changed meteorological conditions. The horizontal line within the boxes gives the median of observed values, diamonds give the averages, upper and lower sides of the boxes give the interquartile range between the 25th and 75th percentiles and the whiskers give the largest and lowest values.

observed (principally under T-4˚C and T-2˚C), when reduced net primary production was reduced by low temperatures combined with rainfall deficit during winter and early spring. Lower temperatures reduced $CO_2$ assimilation rates and hence the production of biomass, and rainfall deficits led to the development of drought during spring. This water limitation occurring early in the season, strongly impairing biomass production of the grasslands so that the amount of grass on the paddocks remained below the threshold needed for cattle grazing.

SOC stocks were also highly sensitive to changes in air temperature and decreased nearly linearly over the range of temperature changes, with a temperature decrease of −4˚C leading to an increase of 12% (13%) in SOC stocks, to reach 125 (122) tC ha$^{-1}$. Conversely, an increase of air temperature by 4˚C would lead to a reduction of −16% (−14%) of SOC, with C stocks falling to 94 (93) tC ha$^{-1}$ by the end of the simulation period. These losses of SOC resulted from increased SOC decomposition rates under increasing air temperatures despite slightly reduced NPP (Fig 1c).

*Effects of changes in annual precipitation.* GPP, NPP, milk production and required N fertilizer application rates (Fig 2) were all positively correlated with changes in precipitation as the alleviation of water stress enhanced carbon assimilation rates (Fig 2a) and hence biomass formation (Fig 2c). In contrast, even given the fact that more biomass was produced on the paddocks, our simulations showed that SOC (Fig 2f) decreased with increasing precipitation. This resulted principally from the combined effects of: 1) a higher proportion of biomass being exported by grazing (mowing), 2) less water stress which reduced the inputs of dead plant materials (litter) to the soil during drought periods and 3) enhanced decomposition of soil organic carbon (Fig 2b) due to higher amounts of water in the soil and less water limitation of SOC mineralization rates. Under increased precipitation scenarios, N fertilizer applications had to be increased to sustain higher biomass production rates and to compensate for more frequent and increased nitrate leaching rates.

A −25% change of annual precipitation intensified water stress and water limitations, leading to a −14% (−13%) change in C assimilation rate (GPP, Fig 2a), −28% (−37%) in milk-solid

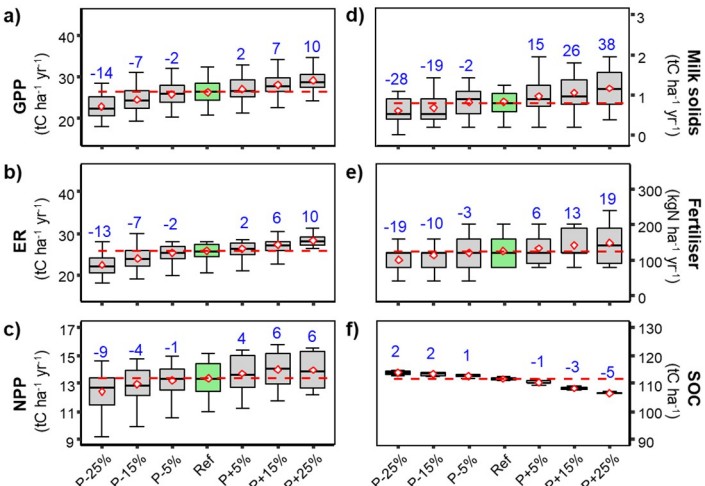

**Fig 2. Box and whiskers charts for modelled variables under a range of different amounts of annual precipitation.** Details as for Fig 1.

production (Fig 2d), −13% (−10%) in ecosystem respiration (ER, Fig 2b) and −11% (−17%) in net primary production (NPP, Fig 2c). With a strong reduction of rainfall (−25% below the baseline), pasture productivity (Fig 2c) was so strongly reduced in some years that above-ground vegetation throughout the year remained too short for cattle grazing, thus allowing no milk production in those years.

Under those dryer conditions, SOC stocks sometimes increased slightly under the combined effects of reduced C exports through grazing even though less biomass was produced and reduced SOC mineralization rates under the inhibition of low soil water contents. Over the full range of changes in precipitation amounts (−25% to +25%), SOC stocks generally only changed marginally with SOC stocks changing by +2% (+5%) and −5% (−4%) between P−25% and P+25%, respectively. SOC changes remained slight because under wetter conditions, enhanced SOC decomposition drove losses of SOC, but greater biomass productivity brought more new C into the system counterbalancing SOC losses.

*Effects of changes in atmospheric $CO_2$ concentration.* All modelled variables except SOC were positively correlated with changes in atmospheric $CO_2$ concentration mostly due to enhanced carbon assimilation rates and hence biomass growth. Higher $CO_2$ concentration led to enhanced GPP (Fig 3a) and, although a proportion of the extra fixed C was lost in increased ecosystem respiration (Fig 3b), both grasslands productivity (NPP, Fig 3c) and resultant milk-solid production (Fig 3d) increased. The increased biomass production then also required higher inputs of nitrogen fertilizers to sustain increased plant needs (Fig 3e). Even though increased atmospheric $CO_2$ concentration increased C flows into the systems, we modelled small losses of SOC. These slight carbon losses from the soil despite increased photosynthetic carbon gain mostly resulted from larger proportions of C removed in milk solids and animal respiration that led to reduced litter inputs to the soil. Under higher atmospheric $CO_2$ concentration, SOC mineralization rates were also slightly enhanced as water stress was reduced through increased water use efficiency that maintained more favorable soil-water contents (Fig 3f).

Doubling $CO_2$ concentration from the baseline of 380 ppm led to important changes in modelled variables of both managed grasslands. The most important changes were found for milk-solid production and N fertilizer application rates that changed by +91% (+87%) and +65% (+70%), respectively. GPP increased by +27% (+30%), ER by +24% (19%) and NPP by

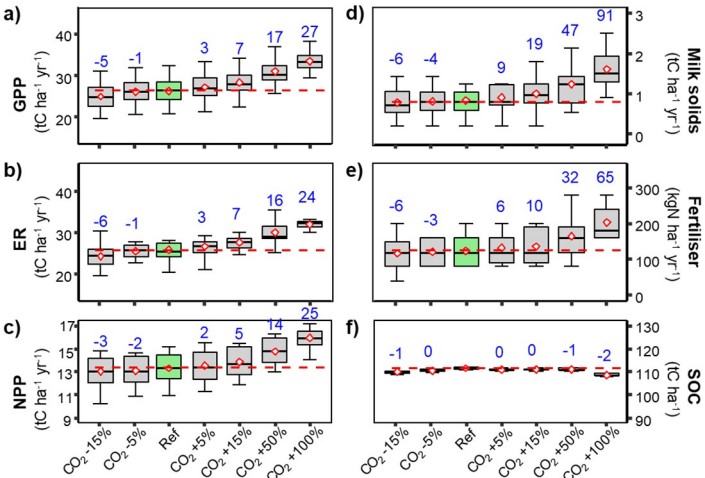

**Fig 3. Box and whiskers charts for modelled variables under a range of different atmospheric $CO_2$ concentrations.** Details as for Fig 1.

+25% (+28%) while SOC decreased, but only by −1% (−3%). The changes in SOC being relatively small, compared to the important changes in other modelled variables highlighted the important effect of management on SOC dynamics in highly managed grasslands and the need to carefully account for all the different components of the system to fully understand how the carbon flows are affected by climate perturbations.

## Climate-driven changes in grasslands productivity and SOC

**Long term responses of modelled variables to climate change scenarios.** Trends in the responses of the different variables generally intensified over time and with RCP. All modelled fluxes showed increasing trends, but at the expense of decreasing SOC. Mean GPP in the H period was 24 (24) tC ha$^{-1}$ yr$^{-1}$ and increased by 11% (12%), 14% (15%) and 28% (30%) in FF under RCP 2.6, RCP 4.5 and RCP 8.5, respectively (Fig 4a). Under the different climate change scenarios, ER (Fig 4b) increased in line with changes in GPP, increasing from 9% (5%) under RCP 4.5 in NF to 30% (21%) under RCP 8.5 in FF.

Grassland productivity (NPP, Fig 4c) and milk-solid production (Fig 4d) significantly increased along with required N fertilizer application rates (Fig 4e). We generally found the largest changes under RCP 8.5 in the FF horizon with increases of 39% (41%), 104% (128%) and 53% (87%) for NPP, milk solids and N fertilizer application rates, respectively.

SOC generally changed very little under the compensating influences of the different climate-change aspects, and the small observed differences were generally small. Under the three RCP scenarios, SOC decreased over all time slices compared to the baseline SOC (Fig 4f). For the grazed paddock, the largest SOC decrease was found to be −5%, under RCP 8.5 in FF, followed by RCP 2.6 and RCP 4.5 with −2%. The results also indicated that SOC changes in IF were slightly decreased by −1%, −1% and −2% under RCP 8.5, RCP 4.5 and RCP 2.6, respectively (Fig 4f).

**Importance of the $CO_2$ fertilization effect on model simulations.** Our simulations highlighted the importance of $CO_2$ fertilization on the potential responses of the modelled variables. Under constant $CO_2$ concentration, all modelled variables showed decreasing trends that intensified toward the end of the century (Fig 5). Mean GPP would at first increase by 6% (7%), 1% (2%) and 7% (8%) in NF under RCP 2.6, RCP 4.5 and RCP 8.5, respectively (Fig 5a) which

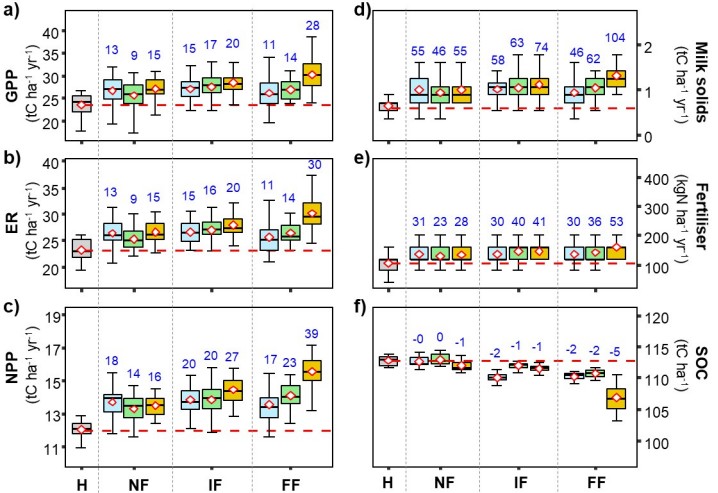

**Fig 4. Box and whiskers charts for modelled GPP a), ER b), NPP c), Milk-solid production d), required N fertilizer application rates e) and SOC stocks f) under different future climate scenarios for the grazed grassland site.** Model runs were done under different climate-change projections according to RCP 2.6 (blue), RCP 4.5 (green) and RCP 8.5 (orange). Driving variables included their respective atmospheric $CO_2$ concentration dynamics. Horizontal red dashed line and red numbers are the mean of the variable for the Historical period and blue numbers give the percent change of the annual variables means for the different scenarios and time horizons.

is considerably lower than the observed increases when increases in $CO_2$ concentration over the same period were also included. Under constant $CO_2$, changes in GPP would stabilize in IF at values close to those found in H and would be reduced in FF by −5% (−4%) and −14% (−14%) under RCP 4.5 and RCP 8.5, respectively. Under the different RCP scenarios, ER (Fig 5b) changed in line with changes in GPP. ER first increased on average from 3% to 7% (0% to 4%) in NF and decreased from 2% to −7% (−1% to −8%) in FF, between the different scenarios.

Changes in grasslands productivities (NPP, Fig 5c) were positive under all RCP scenarios in NF and IF periods but were equal to zero and even negative in FF under RCP 4.5 and RCP 8.5, respectively. In all periods and scenarios, changes in NPP under constant $CO_2$ were much

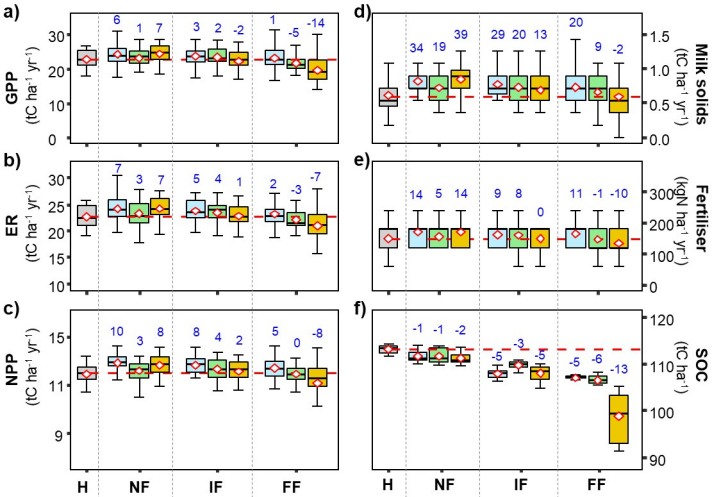

**Fig 5. Box and whiskers charts for modelled responses of grazed grassland under constant atmospheric $CO_2$ concentration (380 ppm).** Other information as for Fig 4.

lower than changes reported with increasing atmospheric $CO_2$ concentrations (Fig 4c). Milk production (Fig 5d) and required N fertilizer application rates (Fig 5e), increased in NF and IF and were then reduced in FF. The trends and response to RCPs was similar for simulations with constant and increasing $CO_2$, but all changes were substantially lower under constant than increasing $CO_2$ over the 21st century. We found that changes were negative under RCP 8.5 in the FF horizon with decreases of −8% (−5%), −2% (0%) and −10% (−7%) for NPP, milk solids and N fertilizer application, respectively. It is also very important to note that under RCP 8.5 and with constant atmospheric $CO_2$ concentration, even if average milk-solid production is only reduced by −2% (0%) in the FF period, there is a large variability and in some years milk production would be reduced to zero. In years for which milk production was reduced to zero, we found that meteorological conditions were unsuitable for plant growth and hence biomass amounts were not sufficient for animal grazing.

Under the three RCP scenarios and with constant $CO_2$, SOC decreased in all time slices compared to H period (Fig 5f). The largest SOC decrease of −13% (−14%), was found under RCP 8.5 in FF, followed by RCP 4.5 and RCP 2.6 with −6% (−6%) and −5% (−7%), respectively. The results also indicated that SOC stocks in IF were slightly decreased by −5% (−6%), −3% (−5%) and −5% (−6%) under RCP 8.5, RCP 4.5 and RCP 2.6, respectively (Figs 5 f and 6). It means that SOC losses were twice as large if the expected increasing $CO_2$ concentrations in the scenarios were not included in the simulations (Fig 6). This is because of the lower biomass production under a drier and warmer climate, causing lower C inputs in the system and greater SOC decomposition rates. That is why the SOC would decrease over time in the 21st century.

It was also interesting to note that simulations testing the sensitivity of modelled variables to factorial changes in climate input showed that increasing $CO_2$ seemed to have very little effect (Fig 3), but that with RCP scenarios, $CO_2$ was much more effective in alleviating any adverse effects of climatic changes on SOC (Figs 4–6).

**Effects of climate extremes on simulations.** We used the correlations between 6 selected indices from ETCCDI (Table 2) and various modelled variables to understand the dependence of various characteristics of the modelled system on the different climatic extremes (Fig 7). Positive correlations between the modelled variables and the ETCCDI indices imply that increases in the indices under the climate change scenarios lead to increases in the modelled variables.

Correlations (both positives and negatives) between climate extreme indices and modelled variable increased as we moved from the optimistic RCP 2.6 scenario to the more pessimistic RCP 8.5 scenario. This suggests that climate extremes play an increasingly important role in the behavior of managed grassland ecosystems. There were also large differences in respective correlations between simulations using constant or increasing atmospheric $CO_2$ concentrations. This highlights the important role of the $CO_2$ fertilization effect in our simulations of managed grasslands productivity, and specifically their responses and adaptability to extreme climatic events. The very strong negative effects of grasslands responses to temperature and water

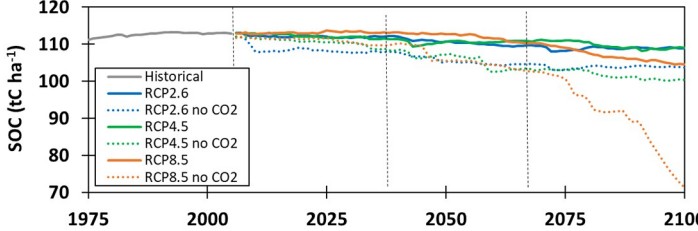

**Fig 6. Time series of modelled soil organic carbon (SOC) stocks modelled under RCP 2.6, RCP 4.5 and RCP 8.5 with and without $CO_2$ fertilization for the grazed paddock.**

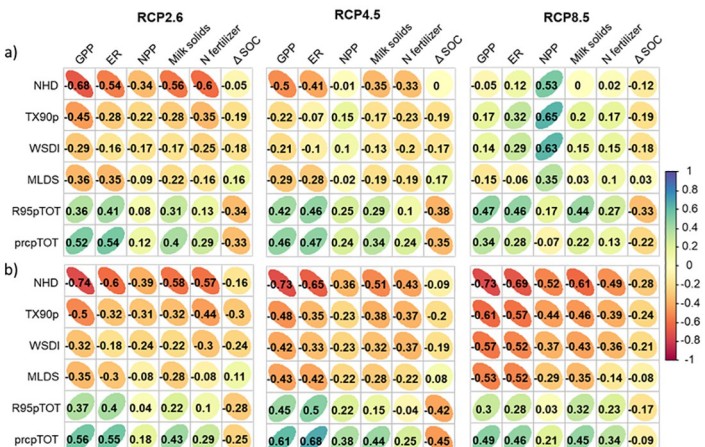

**Fig 7. Correlation matrixes between the six selected ETCCDI climate indices modelled GPP, ER, NPP, Milk solids, required N fertilizer application rates and change in SOC (ΔSOC) under RCP 2.6, RCP 4.5 and RCP 8.5 for grazed pastures with increasing (a) and constant (b) CO$_2$ concentrations.** Numbers give the correlation coefficients and colors indicate correlation levels, going from red for negative to blue for positive correlations.

extremes, observed in Fig 7b, especially under RCP 8.5, were not as strong and sometime even reversed direction when the variations in atmospheric CO$_2$ were included in the simulations.

For example, in our simulations with increasing CO$_2$, increases in the number of hot days (NHD) was highly negatively correlated with GPP (Fig 7a) under RCP 2.6 (−0.68), the correlation was weaker but remained negative under RCP 4.5 (−0.50) and became almost zero under RCP 8.5 (-0.05). In contrast, when the atmospheric CO$_2$ concentration was kept constant in the simulations, the correlations were even stronger and remained equally strong (-0.73) under all three climate change scenarios (Fig 7b).

Changes in SOC were negatively correlated with all ETCCDI indices, except MLDS, and the pattern was the same for all RCPs. and were more correlated with those referring to temperature (NHD, TX90p and WSDI) than water (R95pTOT and prcpTOT). This indicates that increases in temperature and precipitations would have a negative effect on SOC, through the increase of the mineralization rates while increasing drought periods would likely increase SOC by increasing litter inputs and reducing mineralization rates.

## Discussion

### Comparing responses of grazed and mowed systems

Soil organic carbon stocks were significantly different between the mowed and grazed systems, with higher SOC under grazing (112 tC ha$^{-1}$) than under mowing (108 tC ha$^{-1}$). This is explained by the fact that under the mowed (cut and carry) system 95% of all biomass is exported offsite after each mowing event while in grazed systems, about 25% of ingested biomass is returned to the soil in dung and urine [27, 31]. Apart from this difference in carbon stocks between the two systems, both responded very similarly to changes in all environmental variables.

### Model responses to factorial changes in climate variables

The model sensitivities to individual changes in key climate variables (air temperature, precipitation and atmospheric CO$_2$ concentration) provided important insights into different ecophysiological responses to climate change and key vulnerabilities. Temperatures have an important effect on seasonal patterns of growth rates of temperate grasslands [38, 46]. Low

temperatures during winter can limit biomass production, while high summer temperatures can also be stressful, especially if rainfall is insufficient to prevent water availability to become limiting [46, 47]. Generally, most of the modelled variables (Fig 1) were more responsive to reduced air temperature than its increase, suggesting that current temperatures at Lusignan are slightly lower than optimal for pasture productivity. SOC decreased across the range of temperatures, with carbon gain and the proportion exported showing only small changes, but organic matter decomposition being stimulated across the range of modelled temperatures.

Across the range of changes in precipitation, annual GPP was positively related to changes in precipitation, indicating the important control of water availability on productivity at this site. This greater carbon gain was then also reflected in trends in milk production, ER and NPP (Fig 2). These modelled fluxes were quite sensitive to reduction in rainfall, SOC stocks, however, were reduced under increased precipitation and changed little when precipitation was reduced (Fig 2f). The SOC trend observed for simulations with increased rainfall is related to greater grazing biomass off-take over the drier summer months (as shown in detail by [37]). Even though more carbon entered the system with higher precipitation, a larger proportion of the extra biomass could be exported, leaving similar absolute amounts of carbon reaching the soil for organic-matter formation. With the soil remaining moist for longer, decomposition was stimulated, leading to an overall carbon loss. The plateauing of carbon stocks under reduced precipitation was due to lower assimilation rates, hence lower biomass production, counteracted by reduced mineralization of SOC in the drier soil.

Increase in atmospheric $CO_2$ concentration, known as the $CO_2$ fertilization effect led to important increases in grasslands productivity [48]. Apart from changes in SOC and evapo-transpiration (data not shown), all other variables showed positive responses to changes in of atmospheric $CO_2$ (Fig 3). Evapotranspiration was reduced due to partial stomatal closure under higher $CO_2$. Higher $CO_2$ concentration tended to enhance photosynthesis rates (GPP, Fig 3a), (ER, Fig 3b) and consequently grasslands productivities (NPP, Fig 3c). That also necessitated increased nitrogen-fertilizer application rates (Fig 3e) to prevent nutrient limitations. While other nutrients, such as phosphorus, were not modelled in the present work, they could also become limiting under sustained exports in animal feed. These changes in carbon gain led particularly strong responses in milk production (Fig 3d). Milk production responded particularly strongly to increasing $CO_2$ because elevated $CO_2$ not only allowed more carbon to be fixed by the pasture but elevated $CO_2$ also reduced water losses, thereby minimizing summer droughts and allowed a greater proportion of fixed carbon to be grazed and exported from the site. Increasing $CO_2$, this had a similar effect on pasture productivity as increasing precipitation.

Even though grasslands became more productive with more $CO_2$ in the atmosphere, SOC stocks did not increase with increasing $CO_2$. This was principally due to higher proportional biomass removal that limited inputs of organic carbon into the soil and enhanced SOC mineralization rates with lower water limitation because of reduced water losses with partial stomatal closure under higher $CO_2$. Our results are consistent with findings from FACE (Free Air $CO_2$ Enrichment) experiments reporting increased GPP by 20–50% for doubling $CO_2$ concentration for different $C_3$ plant species [24, 49] compared to CenW modelled GPP increases of 27% and 30% for the grazed and mowed paddocks, respectively.

## Long-term simulation of grasslands productivity and SOC dynamics under different RCP scenarios

In addition to sensitivity tests to factorial changes in meteorological variables, we also ran simulations with weather data based on 3 different RCP scenarios for which all meteorological variables were varied together (e.g. Figs 4 and 5).

When the managed grasslands were modelled under the RCP climate forcing with varying $CO_2$ levels according to the considered RCP (Fig 4), all modeled variables responded significantly to climate change in most periods and all RCP scenarios. We observed an increasing trend for all modelled variables (except for SOC) with higher magnitude under RCP 8.5 than RCP 4.5 and then RCP 2.6 and as the 21st century progress. It showed that under future climatic conditions, the productivity of managed grasslands could increase significantly because of higher biomass production due to improved photosynthesis rates and water use efficiency. However, it would require more N fertilizer inputs to meet the plants' needs [50–52] which would be detrimental as grasslands would be likely to lose more N through leaching and $N_2O$ emissions. In addition, we observed that even if CenW modelled an increase in grass production, SOC stocks would be substantially reduced. On one hand, photosynthesis rates and water use efficiencies would be improved and hence more biomass produced. However, on the other hand, more grass could be removed from the paddocks, and higher temperatures could enhance SOC mineralization rate. If N fertiliser application rates were maintained at baseline levels, available N resources would be progressively diminished by N exports, so that grassland productivity would be increasingly compromised by N limitations. The extra fertiliser needs were particularly strong under the RCP 8.5 scenario. This suggests that additional nitrogen fertiliser applications would be required to maintain/increase grassland productivity. Although not modelled in the current study, it is likely that similar limitations would also occur for phosphorus.

The potential temporal responses of SOC to future climatic conditions in the different periods are shown in Fig 6. The climatic conditions under RCP 8.5 scenario would have more negative effects than RCP 4.5 on SOC by the end of the century. The most sever adverse effects were more pronounced under FF than in NF and IF. This is because the effects of $CO_2$ fertilization and climatic variables dominated over the initial period to stimulate biomass production and organic carbon inputs, but more extreme temperatures and water stress developed in FF, limiting biomass production and hence inputs of litter into the soil. These reduced inputs combined with higher temperatures that enhanced mineralization to reduce overall SOC stocks. Average losses of SOC in FF under the 3 RCPs were similar to those found for three managed grassland sites simulation with the PaSim 2.5 model [53].

In brief, elevated $CO_2$ concentration in the atmosphere had the dual effect of increasing photosynthesis, hence favoring biomass production, while reducing stomatal conductance and thus water losses through transpiration [54]. Together, that increase water-use efficiency [23]. Our results on the changes in GPP under future climate are consistent with the findings from the modelling study of [55]. In line with our findings, several Free-Air Carbon dioxide Enrichment (FACE) experiments setup on grasslands reported an increase of productivity induced by the increase of $CO_2$ in the atmosphere [56–58]. However, changes in grassland productivity are also affected by other factors that could offset the benefits from $CO_2$ fertilization alone such as the climate, principally throughout increased temperatures (heat stress) or/and reduced precipitations (water limitations/drought) [59] and nitrogen limitation [60–63].

Damage to grasslands after mowing/grazing events was included for in the CenW simulations [64] but grazing could also have impacts on soil physical, hydrological and thermal properties. Climate extremes could also have additional direct effects on livestock health [65] that were not included in the current assessment. Climate change could affect grasslands biomass production (yield) but it is also expected to affect livestock production by changing the nutritional value of the vegetation [66] and the biological composition (mix of species) of the swards [65, 67–69]. For instance, [68] found that increasing atmospheric $CO_2$ concentration tended to decrease forage nitrogen and protein contents.

To adapt farm management practices, a first step would need to be to specify a goal (such as to increase SOC stocks or maintain/increase productivity) as there are always tradeoffs to take into account [37]. For example, if the overall goal were to reduce the applications rates of N fertilizer, a farmer could plant a sward with a greater proportion of nitrogen-fixing legumes but this could come at the expense of reduced productivity as legumes are less productive than grass species. The system could then be studied in detail with a model like CenW to clearly identify unavoidable trade-offs and define optimal pathways to adaptation, but such an analysis is out of the scope of this paper.

## Effects of $CO_2$ fertilization on modelled variables in the 21$^{st}$ century

When the two grassland systems were modelled with the different RCP climate forcing with constant atmospheric $CO_2$ concentration of 350 ppm, their responses (Fig 5) were very different from those obtained when increases in $CO_2$ concentration was also included in the simulations (Fig 4). Most changes in modelled variables under RCP 2.6 and RCP 4.5 were not significant over the coming century (data not shown), but SOC changes compared to the H period were significant for all RCPs and time horizons (Figs 5f and 6). Without the $CO_2$ fertilization effect, SOC was generally reduced in the simulations. Such losses were not found when the expected $CO_2$ concentration increases were included in the simulations. This is because increasing $CO_2$ concentration could increase photosynthesis and water use efficiency, leading to higher biomass production and hence promoting soil carbon accumulation.

Our results are also in agreement with experiments that showed that a 3–4˚C temperature increase could counterbalance the $CO_2$ fertilization effect on grassland productivity [70]. Furthermore, we also found that elevated $CO_2$ could reduce the grassland sensitivity to low precipitation by increasing plant water use efficiency and thereby reducing the depletion in soil moisture content [71, 72].

Studies [73, 74]. found that changes in precipitation were an important factor driving temperate ecosystems productivity responses to climate change. Generally, increased precipitation led to higher biomass production through improved soil water availability for plants and in contrast, showed that precipitation decrease reduced plants biomass production. This was also observed in this study in the sensitivity to climate factors for the Lusignan site. However, under the different RCPs, expected annual amounts of precipitation at our study changed only marginally (Table 1 and Figs 1 and 2e, 2f in S1 File) and for simulations with constant $CO_2$ levels, temperature changes were the most important driving variable of ecosystem and SOC dynamics. Much greater decreases in SOC are expected under the RCP 8.5 scenario than other scenarios (Fig 6). Not surprisingly, these differences were largely caused by large temperature increases that were much higher under the RCP 8.5 scenario than under the other scenarios (Table 1 and Fig 2 in S1 File). Simulations under those conditions resulted in increased C release and a decrease of C stocks. Further analyzes showed that if the $CO_2$ fertilization effect had not been included in the model runs, water and temperature stresses would have heavily impaired pasture growth in some years (almost killing all vegetation in the paddocks). This would have led to larger losses of SOC [75–79].

In our simulations, elevated $CO_2$ concentration made the dominant contribution to the projected grassland ecosystems productivity and their SOC dynamics, given the fact that we found important modification in projected modelled $CO_2$ and water fluxes, productivity, management intensities and SOC when the effect of increasing $CO_2$ was excluded (Figs 3–5). Our findings are consistent with experimental findings from French grassland sites [55] and would tend to confirm that the biochemical basis of the CenW model is correct.

## Effects of climate extremes on modelled grasslands

Model projections and observational data confirmed that climate extremes are likely to increase with climate change [18, 20, 80]. Applying these climate extremes to our modelled grassland, we showed that the grassland system modelled with and without the $CO_2$ fertilization effect responded very similarly to extreme events.

With the effect of increasing $CO_2$ included in the simulations (Fig 7a), GPP and milk production were negatively correlated with the three temperature indices (NHD, TX90p and WSDI) under RCP 2.6 and RCP 4.5, indicating a reduction of the photosynthesis rate and milk production with the increase of high extreme temperatures while they were positively correlated under RCP 8.5. At the same time, ER was positively correlated with the temperature indices, even highly correlated under RCP 8.5. It indicated that the increase in temperature extremes increased ecosystem respiration rates and enhanced losses of C toward the atmosphere.

Under RCP 8.5 scenario, there was a clear difference in the correlations of GPP, ER, NPP and milk production with the climate indices for the simulations with and without $CO_2$ fertilization. In the simulations with increasing $CO_2$ concentrations, there were positive correlations between the different modelled variables and the increase of ETCCDI indices throughout the 21st century. This indicates that the increase of periods with higher temperature and with rainfall deficit would lead to higher GPP, ER and productivity. It indicated that under higher $CO_2$ concentrations, increasing temperature could stimulate growth while water stress was prevented by the higher water-use efficiency that could be achieved under elevated $CO_2$. For simulations with constant $CO_2$, the correlations were negative, indicating that increases in temperature were detrimental to productivity. In these simulations with constant $CO_2$, increasing temperature led to worsening water shortages, which became the dominant driver of productivity changes. Our simulations under future possible climatic conditions and elevated $CO_2$ concentration agree with the study of [81] who found in a manipulative climate study that elevated $CO_2$ could mitigate the effects of extreme droughts and heat waves on ecosystem net carbon uptake. Thus, this confirmed the strong effect of $CO_2$ fertilization on the carbon and fluxes and grasslands resilience to extreme climatic events (heat waves and droughts).

Interestingly, our results shows that changes in soil carbon stocks (ΔSOC) were generally negatively correlated with the six climate-change indices under all 3 climate change scenarios. This shows that extremely high temperatures and precipitation (high rainfall and water deficit) events generally tend to increase soil C losses. Our results are consistent with the finding of [82] who showed that the carbon sequestration potential of grasslands may me strongly impaired by extremes and more frequent drought events.

In conclusion, our study gave an assessment of possible future changes in productivity and C and water fluxes of managed grasslands at a French study site, and the consequences for management intensity and soil carbon-stock dynamics under different climate change projections. More specifically, it showed that modelled productivity increased in response to future change in meteorological conditions, mainly due to the fertilization effect of rising $CO_2$, and led to a possible intensification of grassland management. With the current trajectory of GHG emissions, RCP 8.5 scenario seems to be the most realistic one and the scenario having the most effects on modelled variables by the end of the 21st century. In the FF period, our simulations showed that climate conditions on some years would likely be unsuitable for the pastoral systems that we studied but on average, we simulated an increase in biomass and milk production and losses of SOC. In addition, under current management, more N fertilizer would need to be applied to take advantage of the boost in biomass production and maintaining productivity.

However, our simulations clearly showed that targeting productivity is likely causing a net loss of soil carbon, accelerating along the 21st century as climate change become more severe and that soil carbon stocks in highly productive grasslands could only be increased through changing management practices and reducing C exports from the sites. Losses in soil carbon were attributed principally to increased temperature and decreased carbon inputs to the soil. Activities such as harvesting/grazing plant biomass, through the removal of aboveground biomass significantly decrease the amount of carbon that would have contributed to soil organic carbon formation. Analyzes of extreme meteorological events also showed the importance of the $CO_2$ fertilization effect on the behavior of modelled grazed grasslands. The six selected ETCCDI indices showed that increases in climate extremes generally led to losses of soil carbon.

## Supporting information

**S1 File. Additional information and corresponding figures for the mowed grassland site.** (PDF)

**S1 Graphical abstract.** (TIF)

## Acknowledgments

We would like to thank the National Research Infrastructure "Agro-écosystèmes, Cycles Bio-géochimique et Biodiversité" (ACBB) for their support in field experiment and the database. We are deeply indebted to Christophe de Berranger, Marie-Laure Decau and Nicolas Mascher for their substantial technical assistance.

## Author Contributions

**Conceptualization:** N. J. B. Puche, M. U. F. Kirschbaum, Abad Chabbi.

**Data curation:** N. J. B. Puche.

**Formal analysis:** N. J. B. Puche.

**Funding acquisition:** Abad Chabbi.

**Investigation:** N. J. B. Puche, M. U. F. Kirschbaum, Abad Chabbi.

**Methodology:** N. J. B. Puche, M. U. F. Kirschbaum, N. Viovy, Abad Chabbi.

**Project administration:** Abad Chabbi.

**Resources:** Abad Chabbi.

**Software:** M. U. F. Kirschbaum.

**Supervision:** N. Viovy, Abad Chabbi.

**Validation:** N. J. B. Puche, M. U. F. Kirschbaum, N. Viovy.

**Writing – original draft:** N. J. B. Puche, Abad Chabbi.

**Writing – review & editing:** M. U. F. Kirschbaum, N. Viovy, Abad Chabbi.

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
