## [Decision Letter · Decision Letter 0]

2 Aug 2022

PONE-D-22-17085Potential impacts of climate change on the productivity and soil carbon stocks of managed grasslandsPLOS ONE

Dear Dr. Chabbi,

Thank you for submitting your manuscript to PLOS ONE. After careful consideration, we feel that it has merit but does not fully meet PLOS ONE’s publication criteria as it currently stands. Therefore, we invite you to submit a revised version of the manuscript that addresses the points raised during the review process.

ACADEMIC EDITOR: The study is interesting while the manuscript has some problems as suggested by the reviewers. The authors should respond to the comments of the reviewers one by one and revise the manuscript accordingly. The revised manuscript would be sent to the reviewers for further reviewing.

We look forward to receiving your revised manuscript.

Kind regards,

Jian Liu

Academic Editor

PLOS ONE

Journal Requirements:

2. We note that Graphical Abstract in your submission contain map/satellite images which may be copyrighted. All PLOS content is published under the Creative Commons Attribution License (CC BY 4.0), which means that the manuscript, images, and Supporting Information files will be freely available online, and any third party is permitted to access, download, copy, distribute, and use these materials in any way, even commercially, with proper attribution. For these reasons, we cannot publish previously copyrighted maps or satellite images created using proprietary data, such as Google software (Google Maps, Street View, and Earth). For more information, see our copyright guidelines: http://journals.plos.org/plosone/s/licenses-and-copyright.

a) You may seek permission from the original copyright holder of Graphical Abstract to publish the content specifically under the CC BY 4.0 license.  

Reviewers' comments:

Reviewer's Responses to Questions

**Comments to the Author**

1. Is the manuscript technically sound, and do the data support the conclusions?

Reviewer #1: Yes

Reviewer #2: Yes

Reviewer #3: Yes

2. Has the statistical analysis been performed appropriately and rigorously? 

Reviewer #1: Yes

Reviewer #2: Yes

Reviewer #3: No

3. Have the authors made all data underlying the findings in their manuscript fully available?

Reviewer #1: Yes

Reviewer #2: Yes

Reviewer #3: No

4. Is the manuscript presented in an intelligible fashion and written in standard English?

Reviewer #1: Yes

Reviewer #2: Yes

Reviewer #3: Yes

5. Review Comments to the Author

Reviewer #1: The manuscript by Puche et al. investigate the impacts of the potential effects of future climatic conditions on productivity and soil organic carbon (SOC) stocks of mowed and rotationally grazed grasslands in France. They used the CenW ecosystem model to simulate carbon, water, and nitrogen cycles in response to changes in environmental drivers and management practices. The article addresses the importance of tailoring farming practices to achieve the dual goals of maintaining agricultural production while safeguarding soil C stocks. This manuscript is an interesting story. However, I do not consider the article ready for publication in its current state, and suggest a major revision, especially of the introduction, result and discussion section. These descriptions of these sections are too boring and should revise them to make clear. In addition, many sentences in this paper are too long and ambiguous. Please revise the long sentences to make it more clear and readily understandable.

1) The paragraphs of Introduction are too scattered and do not focus on major questions. For example, the first, second and third paragraph should be integrated in one paragraph, and focus on main problem that related to this topic. The Introduction also is not very informative in its current state, not explaining the important background to the reader. Considering my comments, the literature review should focus on effects of climate change on the productivity and soil carbon stocks.

2) The Results are not concise enough and are boring. I suggest that the results should concise and concentrate on major results. Please revise the results to make the results clearer.

3) The 4.2.1, 4.2.2, 4.2.3 in the Discussion section should change to 4.2, 4.3, 4.4. I suggest that there is no need for subheadings in Discussion.

4) The Summary and Conclusion is too long and not concise enough. Please revise this section to one paragraph and summarize the main results.

5) The section of 2.2.3. Climate change scenarios is too long and should revise this section to make it clear and articulate.

6) Please check the tense and voice in the manuscript.

7) Please check the punctuation, format and spacing throughout the manuscript.

MINOR COMMENTS:

Line 80-84: This sentence is ambiguous and redundant. Please revise it.

Line 115-119: This paragraph is necessary? I think this section is redundant and not strongly related to other parts. Please revise this section.

Line 127-129: Please revise this sentence and this sentence is chaos.

Line 243: Revise this sentence

Line 355-364: These two paragraphs are too boring and it is strange to introduce the invalid results. Delete these sections. Maybe these sections should merge into the result below.

Line 365-586: The results is too long and boring. Please revise these results.

Line 598: “112 tC ha-1 yr-1” or “112 t C ha-1 yr-1”? The same format of units is same below.

Line 613-614: This sentence is strange and change the sentence structure.

Line 664-665: This sentence is strange and revise it.

Reviewer #2: This is a very interesting study, which is very helpful for us to understand the impact of climate change.

The first half of the thesis is written in too much detail, which is more like a dissertation. I suggest that authors simplify some common sense content，such as Line 120 "Grassland ecosystems cover about one quarter of the Earth’s total land area".

Materials and methods should also be modified more succinctly, and some details of model settings can be put in supplementary materials.

Reviewer #3: This paper touched a very nice topic on effects of grassland on climate change which is important and 100% fits the scope of Journal. But it lacks novelty. They study is basically a model output extraction and visualization. The author just picked an already existing grassland models, simulate some variables under climate change scenarios and Climate sensitivity. Most of the results are just about percentage of change in the output variables he extracted from the model. Considering the fact that the author considered one GCM which is not enough, these outcomes are associated with a lot of uncertainty. The paper has a good base for a nice study development for example by adding spatial dimension or by adding more insights on possible management strategies or deriving components of grassland biomass changes. However, in the current status, the study lacks novelty. Although the writing of the paper was not too bad, but it still requires to be correct by a native. Some of the sentences were not clear to me (below comment). Additionally, some minor improvement is required which was not a minor problem in my opinion if the author justify the novelty of the work.

Details comment.

Lines 144: do you mean that “This study has been based on previous experiments”, the word elsewhere sound a bit abstract

Lines 148-149. The reference citation does not seem to be consistent with the journal format. Please move the website to the bibliography

Line 150: please be more specific with “over the study period”: which study period are you talking about.

Line 156: I do not understand this. Four blocks of five 0.4 ha plot.

Line 166-167. Varied between which values? Please be specific.

Lines 206, how many mowing events per growing seasons? And at which time?

Line 208: what do you mean by “to model the weather conditions”, you do not model weather condition

Lines 215-217. I do not understand this. How did you split five years into two weeks? If this is not the original work of authors, I do not see the necessity to putting material methods of others here.

I do not understand the necessity of figure 1. If it is the original results of the authors, it should go to results section. If it is from previous study, then why reference is missing and why do we need it. I even think putting in supplementary is not necessary

Lines 237-238. I do not understand this. In lines 208 and 209 you mention 2005-2010. In lines 228, you said 2005-2016 and now here 2001-2016.

Lines 241-260, I do not understand this. This section belongs to climate change. Or at least another additional section of scenario analysis I also do not understand why we need to have these scenarios of temperature and precipitation increase/decrease?

The whole lines 263-270 could be removed. These are basic known info. OR at least lines 263-265” “The RCP scenarios……..globalization”. Consider removing this sentence.

271-272. all variables needed for running CenW.

Line 271: explain CNRM2014 experiment.

Lines 277: why did you use abbreviation of RCM when you used this word only once?

Lines 339-343. Some of this paragraph, belongs to result section.

Lines 344-349: some of this belongs to results. Accordingly Tabl2 2 should be in result section.

The whole lines 355-364 is not necessary. This kind of information is usually helpful in report not scientific paper.

Lines 366-367- Please consider removing. Lines 366-372 belongs to methodology. You should explain why these variables important. Just presenting model outputs does not justify this.

Lines 374-383. I would suggest to put results from both grazed and mowed grasslands here. One column for each in fig. 2. The same for figure 3

I do not understand why we need a short section 3.1.1. In section 3.1.2 you still continue talking about the same figure. Why not putting everything in one section.

Lines 385-387. This is expected. Right?

Lines 388-390. I did not understand this part. Please consider rephrasing.

Lines 410-411.”More specifically, in the years for which milk solids productions was reduced to zero (principally under T- 4°C and T-2°C), we carefully analyzed modelled and weather variables time series.” What do you mean? Why were you careful with this scenario and not with the others? You may see you explored the reason for reduced production.

Lines 424-426. What is novel about this result? This is general feedback. Are you expected something different from the model?

Lines 478-481- consider removing. Not necessary.

Lines 486-489: this belongs to discussion.

Lines 508-510. This belongs to methodology.

6. PLOS authors have the option to publish the peer review history of their article (what does this mean?). If published, this will include your full peer review and any attached files.

Reviewer #1: No

Reviewer #2: No

Reviewer #3: No

---

## [Author Response · Author response to Decision Letter 0]

31 Jan 2023

We have included the referrers comments immediately and responded to them individually, indicating exactly how we addressed each concern and describing the changes we have made (e.g., separate file detailing the editors comments point-by-point response).

---

## [Decision Letter · Decision Letter 1]

9 Mar 2023

Potential impacts of climate change on the productivity and soil carbon stocks of managed grasslands

PONE-D-22-17085R1

Dear Dr. Chabbi,

We’re pleased to inform you that your manuscript has been judged scientifically suitable for publication and will be formally accepted for publication once it meets all outstanding technical requirements.

Kind regards,

Jian Liu

Academic Editor

PLOS ONE

Additional Editor Comments (optional):

Reviewers' comments:

Reviewer's Responses to Questions

**Comments to the Author**

1. If the authors have adequately addressed your comments raised in a previous round of review and you feel that this manuscript is now acceptable for publication, you may indicate that here to bypass the “Comments to the Author” section, enter your conflict of interest statement in the “Confidential to Editor” section, and submit your "Accept" recommendation.

Reviewer #2: All comments have been addressed

Reviewer #3: All comments have been addressed

2. Is the manuscript technically sound, and do the data support the conclusions?

Reviewer #2: Yes

Reviewer #3: Yes

3. Has the statistical analysis been performed appropriately and rigorously? 

Reviewer #2: Yes

Reviewer #3: Yes

4. Have the authors made all data underlying the findings in their manuscript fully available?

Reviewer #2: Yes

Reviewer #3: Yes

5. Is the manuscript presented in an intelligible fashion and written in standard English?

Reviewer #2: Yes

Reviewer #3: Yes

6. Review Comments to the Author

Reviewer #2: (No Response)

Reviewer #3: (No Response)

7. PLOS authors have the option to publish the peer review history of their article (what does this mean?). If published, this will include your full peer review and any attached files.

Reviewer #2: No

Reviewer #3: No

---

## [Editor Report · Acceptance letter]

31 Mar 2023

PONE-D-22-17085R1 

Potential impacts of climate change on the productivity and soil carbon stocks of managed grasslands 

Dear Dr. Chabbi:

I'm pleased to inform you that your manuscript has been deemed suitable for publication in PLOS ONE. Congratulations! Your manuscript is now with our production department. 

Kind regards, 

on behalf of

Dr. Jian Liu 

Academic Editor

PLOS ONE